# Transcriptional Regulation of Airway Epithelial Cell Differentiation: Insights into the Notch Pathway and Beyond

**DOI:** 10.3390/ijms241914789

**Published:** 2023-09-30

**Authors:** Guadalupe Cumplido-Laso, Dixan A. Benitez, Sonia Mulero-Navarro, Jose Maria Carvajal-Gonzalez

**Affiliations:** Departamento de Bioquímica, Biología Molecular y Genética, Facultad de Ciencias, Universidad de Extremadura, 06071 Badajoz, Spain; benitezlda@unex.es (D.A.B.); smmulero@unex.es (S.M.-N.)

**Keywords:** airway epithelial cells, multiciliated cells, secretory cells, Notch, transcription factors, airway epithelium

## Abstract

The airway epithelium is a critical component of the respiratory system, serving as a barrier against inhaled pathogens and toxins. It is composed of various cell types, each with specific functions essential to proper airway function. Chronic respiratory diseases can disrupt the cellular composition of the airway epithelium, leading to a decrease in multiciliated cells (MCCs) and an increase in secretory cells (SCs). Basal cells (BCs) have been identified as the primary stem cells in the airway epithelium, capable of self-renewal and differentiation into MCCs and SCs. This review emphasizes the role of transcription factors in the differentiation process from BCs to MCCs and SCs. Recent advancements in single-cell RNA sequencing (scRNAseq) techniques have provided insights into the cellular composition of the airway epithelium, revealing specialized and rare cell types, including neuroendocrine cells, tuft cells, and ionocytes. Understanding the cellular composition and differentiation processes within the airway epithelium is crucial for developing targeted therapies for respiratory diseases. Additionally, the maintenance of BC populations and the involvement of Notch signaling in BC self-renewal and differentiation are discussed. Further research in these areas could provide valuable insights into the mechanisms underlying airway epithelial homeostasis and disease pathogenesis.

## 1. Introduction

The airway epithelium plays a vital role in maintaining optimal respiratory function throughout the airway, from the trachea to the alveoli. It possesses innate immunological properties and acts as a barrier to eliminate inhaled pathogens and toxins, thereby preserving lung homeostasis [1,2]. This epithelium consists of various cell types, each contributing to specific functions essential to proper airway function [3].

Previous studies have established that basal cells (BCs) are the primary stem cells in the respiratory tract, possessing self-renewal capabilities and the capacity to differentiate into both multiciliated cells (MCCs) and secretory cells (SCs) [4]. However, chronic respiratory diseases can disrupt the mucociliary dynamics of the epithelium, leading to a decrease in the number of MCCs and an increase in the number of SCs [5,6]. Additionally, viral infections such as the Severe Acute Respiratory Syndrome Coronavirus 2 (SARS-CoV-2) can produce an alteration in the normal regeneration of the respiratory epithelium, promoting the proliferation of basal cells [7,8]. 

The processes of MCC and SC differentiation require the activation of a transcriptional program that specifies cell fate. Among regulatory pathways, NOTCH signaling provides the differentiation of basal cells into secretory cells types, while its inhibition is required for the differentiation of multiciliated cells [9,10]. 

The generation of multiciliated cells involves a program in which two master transcription regulators participate, GMNC (geminin coiled-coil domain containing) and MCIDAS (multi-ciliate differentiation and DNA synthesis) associated cell cycle protein, under the inhibitory regulatory control of GEMININ and NOTCH [11,12,13]. Their induction of essential multiciliogenesis genes [e.g., *TP73* (tumor protein 73), *MYB* (myeloblastosis), *RFX* (regulatory factor X) and *FOXJ1* (forkhead box J1)] results in downstream centriole amplification and basal body docking. Additionally, ciliogenesis is also accompanied by the up-regulation of multiple axonemal dyneins and other motile cilia proteins [14,15,16,17,18,19]. 

Recent studies have revealed that the NOTCH pathway can switch on the HEY1 (Hairy/enhancer-of-split related with YRPW motif protein 1) transcriptor factor to repress the multiciliated cell gene expression program and acquire a secretory cell precursor identity [20]. Besides HEY1, the differentiation of SCs involves the contribution of several transcription factors, such as SAM-pointed-domain-containing Ets-like factor (SPDEF), FOXA2 (forkhead box A2), TTF-1 (thyroid transcription factor 1), FOXM1 (forkhead box M1), RUNX2 (RUNX family transcription factor), and Krüppel-Like Factors (KLF5 and KLF4), among others [21,22,23,24,25,26,27,28].

Several recent reviews have reported different issues about airway epithelium differentiation [7,29,30,31,32,33]. The present review will focus on providing an overview of airway epithelial differentiation pathways. Specifically, this review aims to emphasize the pivotal role of various transcription factors involved in the differentiation process from BCs to MCCs and SCs within the airway epithelium. Furthermore, we highlight the relevant role of the NOTCH pathway to determine epithelial cell fate. 

Regarding the search strategy to perform this review, we searched PubMed, Scopus, Web of Science and Google Scholar using different keywords and the combinations of each of them. On the one hand, the criteria for study inclusion were (1) human and mice studies; (2) airway epithelial cells; (3) processes involved in airway epithelium differentiation; (4) experimental in vitro models of cell differentiation; (5) publication year from 2002 to 2023; (6) English language; (7) papers with available abstract. On the other hand, exclusion criteria were (1) studies in other species (e.g., zebrafish, xenopus); (2) studies in other cellular models (e.g., skin cells, renal cells); (3) studies focusing on epithelial cells’ development; (4) studies focusing on cancer cells and other respiratory pathology; (5) papers outside the topic.

Two reviewers (G.C.L. and D.A.B.) independently screened the articles for relevance based on titles and abstracts. Then, the full texts of the retrieved studies were evaluated by all the authors. The management and organization of the bibliographic sources were carried out using the computer program Mendeley (www.mendeley.com) (accessed on 10 July 2023). The extracted information and results were analyzed and summarized, and then they were included in this review.

## 2. Cellular Composition of the Airway Epithelium

At the cellular level, the epithelium of the mouse trachea and human airways primarily consists of BCs, MCCs, and SCs (main airway epithelial cells) (Figure 1) [4,34,35].

Recent advancements in single-cell RNA sequencing (scRNA-seq) techniques have provided valuable insights into the cellular composition of both the mouse and the human airway epithelium [36,37,38]. Consequently, data analysis from RNA-seq studies on airway epithelial tissue has unveiled the presence of specialized and/or rare cell types, such as neuroendocrine cells (NECs), brush cells or tuft cells, and ionocytes [39,40,41,42]. Furthermore, scRNA-seq techniques have identified additional intermediate cell populations, including deuterosomal cells and suprabasal or parabasal cells, in both human and mouse airway epithelia [4,29,43].

**Figure 1 ijms-24-14789-f001:**
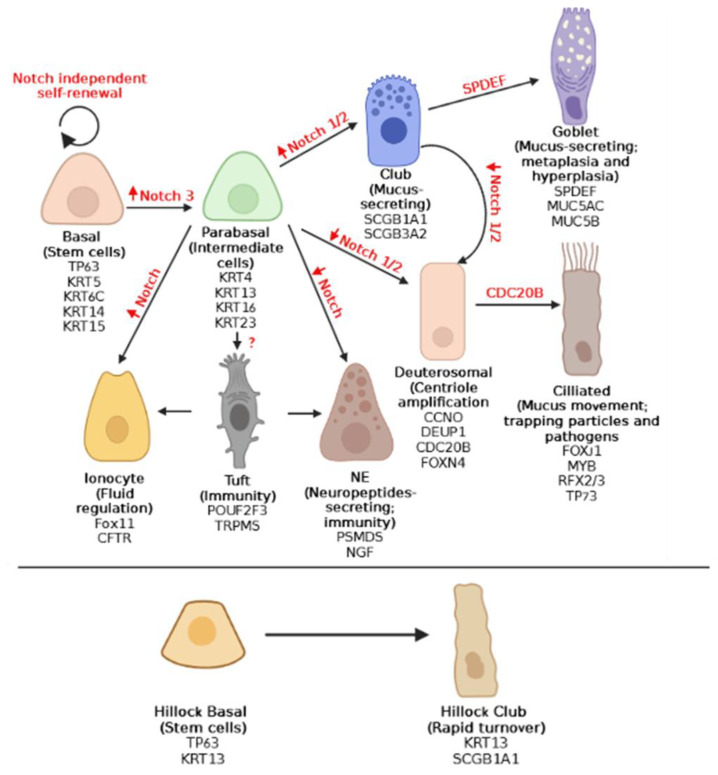
Airway epithelial differentiation pathways. The airway epithelium is mainly composed of MCCs, SCs, and BCs, as well as more rare cell types including ionocytes, NECs, tuft cells, and deuterosomal cells, according to recent human scRNAseq studies [29,44]. BCs have the capacity to self-renew and produce all epithelial cell types [33]. NOTCH pathway can regulate cell differentiation processes, where up- and downward red arrows indicate stimulation and suppression, respectively. Self-renewal basal cells are independent of NOTCH, while the activation of NOTCH signaling will determine epithelial cell fate toward secretory (club and goblet) or ciliated cells, for which the level of NOTCH2 signaling is decisive [9]. Club cells can undergo trans-differentiation into ciliated cells, through NOTCH 1/2 pathway inhibition. For each cell (sub)type, its function and the most important cellular markers are indicated. TP63, KRT5, KRT6C, KRT14 and KRT15 are basal cell markers, while supra-basal cells are distinguished by the expression of KRT4, KRT13, KRT16 and KRT23. The deuterosomal cells express CDC20B, CCNO, FOXN4, and DEUP1, and mature ciliated cells are distinguished by the expression of FOXJ1, MYB, RFX2/3, and TP73. Club cells are characterized by the expression of SCGB1A1, and SCGB3A2, and mature goblet cells express SPDEF, MUC5B, and MUC5AC. Supra-basal cells can also differentiate into ionocytes (FOXI1+ and CFTR+), NECs (PSMD5 and NGF), or tuft cells (POUF2F2+ and TRPM5+). Tuft cells in turn can differentiate either to ionocytes or to neuroendocrine cells. While NOTCH signaling seems to be involved in ionocytes’ and tuft cells’ differentiation, it is still unclear (?) which signaling pathways are involved in NECs’ lineage. Moreover, “Hillocks” TP63+/KRT13+ basal cells give rise to SCGB1A1+/KRT13+ club cells [9]. Created with BioRender.com (accessed on 10 July 2023).

### 2.1. Main Airway Epithelial Cells

In this subsection, the three main airway epithelium cells (BCs, SCs and MCCs) are described in detail.

Firstly, as outlined above, BCs reside in the airway epithelium. They comprise approximately 30% of the airway epithelium and are anchored to the basal lamina through desmosomes [10,45,46,47,48]. Extensive research has demonstrated that BCs serve as the primary stem cells, with remarkable regenerative potential for the airway epithelium, acting as precursors for specialized cell types [4,39,41,47,49]. 

Molecularly, BCs are characterized by the expression of transformation-related protein P63 (TP63), as well as cytoskeletal proteins Keratin 5 (KRT5), KRT6C, KRT14, and KRT15 [45,47,50] (Figure 1). TP63, a member of the p53 family of transcription factors, plays a crucial role in BC development. Studies using mutant mice lacking this protein have shown an absence of BCs in the pseudostratified epithelium of the trachea [51,52]. 

Within the BC population, there are parabasal or suprabasal cells that have detached from the basal lamina to contribute to cell renewal [10]. Previous investigations have confirmed that the transition from basal to suprabasal cells is driven by the endogenous activation of NOTCH3 signaling in murine tracheal and proximal airways (e.g., [43]) (Figure 1). Suprabasal cells are characterized by the expression of KRT4, KRT13, KRT16, and KRT23, and they exhibit lower levels of p63 compared to BCs [29] (Figure 1). These intermediate cells have the potential to differentiate into MCCs, into SCs, or directly into tuft cells [4,10,43,49].

Another group of relevant epithelial cells are SCs, which are interspersed among other cell types in the airway epithelium and play a crucial role in secreting mucus into the respiratory tract lumen, facilitating the trapping of suspended particles [53]. As depicted in Figure 1, SCs encompass a heterogeneous population, including club cells (previously known as Clara cells) and goblet cells [53]. Club cells, which have been extensively studied, exhibit mature states characterized by the presence of the secretoglobins SCGB1A1 and SCGB3A2 [54]. Goblet cells, on the other hand, are molecularly identified through the expression of mucin 5AC (MUC5AC), mucin 5B (MUC5B), and SPDEF [55].

Finally, the third type of main cells of the respiratory epithelium are MCCs. They are specialized airway epithelial cells characterized by the presence of multiple motile cilia on their apical surface. These cells play a crucial role in facilitating the movement of mucus that coats the epithelium, which aids in trapping particles and pathogens [56,57]. Consequently, dysfunction in MCCs can lead to recurrent respiratory infections [58]. As depicted in Figure 1, several molecular markers associated with MCC cells, such as FOXJ1, RFX2, RFX3, MYB, and p73, are instrumental in regulating the expression of numerous proteins involved in MCC differentiation [32].

### 2.2. Intermediate and Specialized and/or Rare Cell Types

Recent studies (e.g., [59]) have identified a population of cells called deuterosomal cells inside the cellular composition and the hierarchy of the airway epithelium, through scRNA-seq, which represent a transitional state during MCCs differentiation (Figure 1). Deuterosomal cells exhibit an intermediate phenotype between SCs and mature MCCs. These cells express specific markers such as CDC20B (cell division cycle 20 homolog B) and DEUP-1 (deuterosome assembly protein 1), which are involved in centriole amplification and the organization of apical microtubules required for cilia formation [12,31]. 

Additionally, deuterosomal cells [60] also express FOXJ1 but display distinct marker expression compared to mature MCCs, including FOXN4 (Forkhead Box N4), CDC20B, DEUP1, and CCNO (cyclin-O) [60,61]. This intermediate cell type has been observed in both human and mouse airway epithelia [59,60]. Furthermore, CCNO, a member of the cyclin protein family required for cell cycle progression, is involved in the early stages of deuterosome-mediated centriole formation [31]. The expression of both CDC20B and CCNO is regulated by MCIDAS (Multicilin (MCI) and IDAS, encoded by MCIDAS), a key regulator of multiciliogenesis [60,62].

Figure 1 also illustrates the presence of tuft cells (also known as Brush cells), which are specialized and/or rare cell types and are involved in immune responses. These cells are characterized by the expression of common markers such as transient receptor potential cation channel subfamily M member 5 (TRPM5) and POU domain class 2 homeobox 3 (POU2F3) [4,63]. The transcription factor POU2F3 plays a crucial role in the formation and differentiation of tuft cells [63]. Additionally, Bukowy-Bieryllo et al. (2022) [29] suggest that tuft cells may serve as progenitors for NECs and ionocytes (as depicted in Figure 1). However, it must be noted that NECs and pulmonary ionocytes can also directly differentiate from BCs through NOTCH signaling [39,41,64].

In regards to the NECs, they constitute approximately 0.5% of all epithelial cells in human airways and express markers such as PSMD5 (Proteasome 26S Subunit, Non-ATPase 5) and NGF (Nerve Growth Factor) [29,65,66]. These innervated epithelial-resident cells play a role in sensing airway activity and secreting neuropeptides to stimulate immune responses [30]. NECs contribute to immune modulation through chemical secretion and are molecularly characterized by the expression of Chromogranin A (CHGA) and calcitonin gene-related peptide (CGRP) [2,4,35,55]. Similar to basal and club cells, a subset of NECs can undergo reprogramming in response to epithelial damage, assuming alternative cell fates and serving as stem cells to facilitate tissue repair [66,67]. 

As for pulmonary ionocytes, these cells exhibit high expression levels of FOXI1 and cystic fibrosis transmembrane conductance regulator (CFTR), suggesting their involvement in the regulation of airway surface liquid (as depicted in Figure 1). The CFTR gene encodes a critical chloride channel that is frequently mutated in cystic fibrosis pathology [35,41]. 

Lastly, Montoro et al. (2018) [39] discovered a novel population of basal progenitor cells, referred to as Hillock basal cells, which express both TP63 and KRT13 (as shown in Figure 1). These Hillock basal cells are known to differentiate into Hillock club cells, exhibiting elevated levels of markers such as Scgb1ab1+ and KRT13+. Although Hillock club cells display high turnover rates, their precise physiological role remains relatively unexplored [39].

Therefore, BCs are the major stem cells of the respiratory tract with the ability to differentiate into ciliated cells, club cells, tuft cells, neuroendocrine cells, and pulmonary ionocytes. For this reason, we will focus this work on reviewing the basal stem cell population maintenance, including both MCC and SC differentiation programs.

## 3. Basal Stem Cells’ Population Maintenance 

As previously indicated, BCs play a crucial role in the development of mammalian airway epithelium, as they possess the ability to self-renew and give rise to all epithelial cell types [33]. NOTCH signaling is highly conserved and plays a relevant role as master regulator of various biological functions. While NOTCH signaling has been implicated in the differentiation of adult airway BCs, their self-renewal appears to be independent of NOTCH [9,10,68]. In addition, Garrido-Jimenez at al. (2021) [69] reported that p53 protein contributes to the self-renewal and differentiation of basal stem cells in mouse and human airway epithelium.

### 3.1. NOTCH Signaling in Airway Epithelium

The mammalian NOTCH family consists of four receptors (NOTCH1-4) and five ligands (delta-like ligand (DLL) 1, 3, 4, and jagged (JAG) 1 and 2) [70]. Human bronchial epithelial cells express all four NOTCH receptors and all NOTCH ligands, with the exception of DLL3 [71]. 

Several studies provide ample evidence that NOTCH3 signaling is involved in the generation of parabasal cells, which subsequently differentiate into secretory or ciliated cells through the activation of the NOTCH1 and NOTCH2 pathways [43,72]. Furthermore, the suppression of NOTCH1–NOTCH2 signaling leads to the expansion of MCCs, while the activation of NOTCH signaling determines the fate of lung epithelial cells towards secretory lineages such as club and goblet cells [32,43,71,72]. Recently, the up-regulation of NOTCH signaling was observed in SARS-CoV-2-infected cells [8]. In cells directly infected by SARS-CoV-2, the NOTCH pathway might promote viral entry, and in addition, excessive NOTCH signaling may promote interleukin 6 (IL-6) and inflammatory pathways that can exacerbate the morbidity and severity of COVID-19 [7].

It must be pointed out that NOTCH2 plays a more central role than NOTCH1 in determining the Clara/ciliated cell fate [9]. Additionally, studies have shown that secretory progenitor cells (club cells) can undergo trans-differentiation into ciliated cells, a process involving NOTCH1–NOTCH2 pathway inhibition [31,33,73]. Rock et al. (2011) [10] demonstrated that BC differentiation in the airways relies on NOTCH signaling, as the inhibition of NOTCH signaling using the gamma-secretase inhibitor DBZ resulted in the absence of intermediate cells responsible for generating ciliated and secretory lineages, leaving only TRP63+ BCs in a pseudostratified epithelium.

### 3.2. p53 Function in BCs' Homeostasis and Differentiation

Recent studies conducted in mouse tracheal epithelial cell culture and human bronchial epithelial cells have highlighted the significance of regulating p53 protein levels in maintaining the self-renewal, differentiation competence, and homeostasis of BCs [69]. In addition, the MDM2/p53 interaction is involved in the regulation of p53 protein levels during BC differentiation, and the disruption of this interaction leads to defects in cell differentiation and alterations in cilia formation [69].

## 4. Transcription Factors Related to Multiciliated Cell Differentiation

As mentioned previously, the differentiation of MCCs, either through NOTCH signaling inhibition in BCs or the trans-differentiation of SCs in the airway epithelium, raises the question of what occurs at the transcription factor level. In the next subsections, the several stages together with trasnscription factors involved in MCC differentation are described in detail. 

### 4.1. Transcriptional Regulators Implicated in Initial Stages of MCC Differentiation 

Previous studies have demonstrated the collaborative role of various members of the GEMININ family in determining the fate of progenitor/stem cells [11]. As depicted in Figure 2, the initial stages of MCC differentiation involve protein interactions, including GEMININ (encoded by GMNN), GEMC1 (Geminin coiled-coil-domain-containing protein 1, encoded by GMNC), MCIDAS, E2F transcription factor (E2F4/5), and DEUP1 [11,12,13].

The expression analysis of these proteins has revealed that in progenitor cells of the airway epithelium, GEMININ is highly expressed, while MCIDAS and GEMC1 are expressed at low levels (Figure 2). Consequently, GEMININ acts as an inhibitor of the transcriptional activation by MCIDAS and GEMC1 [11,12,13]. It has been proposed that GEMININ may prevent the initiation of the multiciliogenesis program until dividing cells exit the cell cycle [12]. Subsequently, GEMININ is down-regulated both transcriptionally and through proteolysis, leading to decreased levels and the disinhibition of GEMC1, which in turn increases its expression [11]. 

GEMC1 interacts with E2F4/5 transcription factors and DEUP1, facilitating MCIDAS transcription [12]. MCIDAS forms complexes with E2F4/5 and DEUP1 (EDM complex), thereby activating genes involved in centriole amplification (cMYB and CCNO) as well as genes necessary for cilia formation (FOXJ1 and RFX), ultimately directing the cell toward multiciliogenesis [11,32,75]. Furthermore, MCIDAS acts on FOXN4 [76], CP110 [77], and p73 [17,78], which also serve as regulators of ciliogenesis. Hence, MCIDAS activates the expression of transcriptional regulators involved in cilia formation, promoting the production of basal bodies for MCC formation [11].

Various observations have highlighted the involvement of additional transcription factors in the process of MCC differentiation. Granhead-like2 (GRHL-2) and the aryl hydrocarbon receptor (AHR) have been identified as transcriptional inducers of MCIDAS [79]. GRHL-2 not only regulates airway cell polarity, barrier function, and differentiation but also acts in parallel with MCIDAS to regulate ciliogenesis [79]. Similarly, AHR transcriptionally activates MCIDAS and CCNO, thereby inducing MCC formation in the airway epithelium [79,80]. 

Cytokinin signaling, as demonstrated by [81], also influences ciliated cell fate. IL-6 and STAT-3 stimulate MCC differentiation by downregulating NOTCH signaling and activating genes such as MCIDAS and FOXJ1 involved in ciliogenesis. Conversely, interleukin 13 (IL-13) inhibits ciliated cell differentiation by suppressing MCIDAS and FOXJ1 expression [81]. 

### 4.2. MCCs Differentiation Stages

After the initial stages of MCC differentiation described above, this process can be categorized into distinct stages based on the expression of various transcription factors and the localization of proteins required for cilia formation (Figure 2). These stages include basal body biogenesis protein synthesis (Stage I), basal body biogenesis (Stage II), the migration and docking of basal bodies in the apical membrane (Stage III), and the generation of motile cilia from the basal bodies (Stage IV) [74]. Transcription factors such as Trp73, MYB, FOXJ1, and RFX2/3, among others, play critical roles in regulating gene expression and orchestrating the differentiation of MCCs in these four stages.

#### 4.2.1. Transcription Factors Implicated in Stages I/II

Transformation-related protein 73 (Trp73) is a member of the p53 family of transcription factors [17,18]. It exists in two isoforms: the activating form (TAp73) and the N-terminally truncated form (DNp73). TAp73 is essential for airway multiciliogenesis, while DNp73 acts as a dominant-negative inhibitor of p53/TAp63/TAp73 [18,78,82,83]. 

The co-expression of p73 and p63 has been observed in a subset of BCs in murine tracheal cells, with p73 acting as a marker for BCs during MCC differentiation [78]. Trp73-deficient mice exhibit the impaired maintenance and differentiation of BCs, defects in basal body docking, axoneme extension, and motility, as well as hyperplasia and loss of the airway epithelium [18,84]. 

Furthermore, p73 is induced by GEMC1 and MCIDAS through the E2F4/5-DEUP1 complexes, contributing to the transcriptional regulation of genes involved in multiciliogenesis [12,16,17,18]. Notably, p73 regulates the expression of key genes such as FOXJ1, RFX2, RFX3, MYB, and miR34bc, which are essential for various stages of MCC differentiation, including centriole amplification and the apical docking of centrioles with axoneme components [74,78,84] (Figure 2).

Another transcriptor factor involved in Stages I/II is c-MYB, which plays a role in promoting the S phase in progenitor cells [19]. In the context of multiciliogenesis in the airway epithelium, MYB expression is initiated after progenitor cells exit the cell cycle [74]. It is involved in centriole amplification, but its activity is turned off once centrioles dock and MCCs mature. 

The inactivation of MYB leads to a failure of complete ciliation in the airways due to impairments in centriole amplification and the expression of FOXJ1, a transcription factor crucial for centriole docking and ciliary motility [74]. MYB is also influenced by NOTCH and MCIDAS signaling and acts upstream of FOXJ1, contributing to centriole amplification [74,85] (Figure 2). Moreover, MYB is required for early multilineage differentiation of airway epithelial cells, specifically in distinct intermediate progenitor cells. These cells, characterized by being MYB+ and p63−, play a crucial role in normal differentiation processes [38]. 

Additionally, studies suggest that CCNO, another protein involved in multiciliogenesis, may promote a cell cycle state that supports centriole amplification and compensates for MYB deletion, enabling multiciliogenesis in specific cell types [58].

#### 4.2.2. Transcription Factors Implicated in Stages III/IV

Cilia formation (apical docking and cilium maturation) occurs in Stages III/IV in which the FOXJ1 and RFX transcription factors play a relevant role (Figure 2).

FOXJ1 is a forkhead box (f-box) transcription factor that plays a central role in the differentiation of ciliated airway epithelial cells [86]. Studies using mouse airway epithelium cells have shown that FOXJ11 expression promotes differentiation during the late stage of ciliogenesis in committed ciliated cells [86]. 

FOXJ1 functions by establishing mechanisms for the docking of basal bodies at the apical membrane and inducing axoneme assembly, contributing to both ciliogenesis in MCCs and the expression of axonemal proteins involved in ciliary motility [15] (Figure 2). FOXJ1-deficient mice studies have revealed that the production of basal bodies is not affected, but their docking at the apical surface of cells is impaired, leading to defects in ciliogenesis and the loss of axonemes in motile multicilia [58,87]. 

FOXJ1 is regulated by TAp73, GEMC1, and MCIDAS during early multiciliogenesis stages (Figure 2), and other transcription factors such as RFX2/3 act as coactivators of FOXJ1 in the differentiation of MCCs [11,87,88,89,90]. The regulatory network of FOXJ1 involves important effector proteins with ciliary roles, identified through cilia-associated expression analysis and gene ontology (GO) analysis, such as PLSCR1 (phospholipid scramblase 1), SSX2IP (SSX Family Member 2 Interacting Protein), ACTN2 (actinin alpha 2), CDC42 (cell division control protein 42 homolog), CFAP206 (cilia and flagella fssociated protein 206), and PIAS4 (protein inhibitor of activated STAT protein 4), among others [15,91,92,93,94,95,96].

The RFX transcription factor family consists of nine members, but only three of them (RFX2, RFX3, and RFX4) have been found to be functionally associated with motile cilia in various tissues [14,15,97,98]. Studies indicate that RFX2/3 and FOXJ1 form a transcriptional complex that specifically regulates genes related to motile cilia [88]. 

RFX3 and RFX2 serve as transcriptional coactivators of FOXJ1 (Figure 2), promoting the expression of genes involved in the differentiation of MCCs [88,89]. Moreover, FOXJ1 can induce the expression of RFX2 and RFX3 in the human airway epithelium, suggesting their involvement in the differentiation of BCs into MCCs [88]. The expression of RFX2 and RFX3 is observed in the late phase of ciliogenesis, indicating their distinct roles in promoting airway ciliogenesis. 

The deletion of both RFX2 and RFX3 results in significant impairment of airway ciliogenesis, leading to a reduction in the number of multiciliated tracheal cells, while individual deletion of RFX3 or RFX2 does not cause ciliary defects in the trachea [97,99]. Notably, RFX3 has been shown to induce the expression of several axonemal dynein genes, including Dnahc9, which encodes a dynein protein associated with human motile respiratory cilia [100]. On the other hand, RFX2 is predominantly expressed in the bronchial epithelium and is involved in the differentiation of ciliated cells from basal progenitor cells during epithelial regeneration [79].

Additionally, there are other regulators involved in MCC differentiation in the airway epithelium, including Transformation/transcription-domain-associated protein (TRRAP) [101], FOXN4 [31,76], Fibronectin type 3 and ankyrin repeat domains 1 (Fank1), JAZF zinc finger 1 (Jazf1) [102], and aryl hydrocarbon receptor (Ahr) [80].

## 5. Transcription Factors Related to Secreted Cell Differentiation 

As depicted in Figure 3, BCs undergo a NOTCH-dependent process to generate multipotent suprabasal cells [71]. These suprabasal cells possess a limited proliferative capacity and give rise to differentiated ciliated or secretory cells upon the activation of NOTCH signaling [71]. In addition, HEY1 transcriptor factor can be activated by the NOTCH pathway to repress the multiciliated cell gene expression program and acquire a secretory cell precursor identity [20]. Among other transcription factors involved in the differentiation of SCs are SPDEF, FOXA2, TTF-1, FOXM1, RUNX2, and KLF5 and KLF4.

### 5.1. NOTCH Signaling and HEY Transcriptor Factor in SC Differentiation

In the process of airway epithelial differentiation, NOTCH3 can prime a subset of BCs to differentiate into club cells, which subsequently have the potential to differentiate into goblet or ciliated cells through the downstream signaling of NOTCH1/NOTCH2. The availability of NOTCH ligands Jag1 (Jagged1) and Jag2 is necessary for this process [43,107]. Specifically, Jag1 has been implicated in the differentiation of human airway epithelial cells toward SCs [70].

Moreover, recent studies have revealed the regulatory role of the NOTCH pathway on two transcription factors, namely HEY1 and MYCL, as illustrated in Figure 3. These transcription factors play crucial roles in determining the fate of intermediate cells during the differentiation of basal stem cells in the airway epithelium [20]. Intermediate cells harbor the expression of HEY1 and MYCL, with MYCL acting as an early driver of MCC differentiation and HEY1 functioning as an early repressor of the MCC fate specifically expressed in SC precursors. Consequently, intermediate cells activated by the NOTCH pathway activate the transcription factor HEY1, leading to the repression of the MCC gene expression program and the acquisition of an SC precursor identity. Conversely, intermediate cells that do not receive active NOTCH signaling trigger a network involving MYCL, GMNC, and MCIDAS, ultimately leading to their differentiation into MCC precursors [20].

### 5.2. Other Transcriptor Regulators Implicated in Goblet Cell Differentiation

In addition to HEY1, the differentiation of SCs involves the contribution of various transcription factors, including SPDEF, FOXA2, TTF-1, FOXM1, RUNX2, and KLF5 and KLF4, among others (Figure 3). 

Specifically, SPDEF is essential for goblet cell differentiation and the production of mucus, particularly the major secreted airway mucin MUC5AC [23,24]. Conversely, FOXA2 acts as a potent inhibitor of goblet cell differentiation in the lung [21,23,24,103]. FOXA2 and TTF-1 (also known as NKX2-1) are critical transcriptional regulators that are selectively expressed in club cells and function to inhibit goblet cell differentiation in the lungs [21,104,105,106]. 

Several studies have shown that SPDEF expression is upregulated in mice and human airway epithelial cells in response to IL-13 or pulmonary allergens [21,97,106,108]. This up-regulation of SPDEF leads to increased expression of genes associated with goblet cell differentiation and mucin biosynthesis, glycosylation, and packaging (Figure 3). 

In contrast, SPDEF inhibits the expression of characteristic genes of club cells in normal bronchiolar epithelium, including FOXA2, TTF1, and genes involved in fluid and electrolyte transport and innate host defense (Figure 3) [21,106,108]. It is important to keep in mind that goblet cells are typically not abundant in the conducting airways of the lung; however, their differentiation can be induced by acute and chronic inflammatory stimuli such as IL-13 and allergens, leading to changes in goblet cell numbers and activity [24].

Additionally, it has been reported that another transcription factor, FOXM1, plays a role in goblet cell differentiation by directly activating SPDEF upon exposure to allergens [25,109]. FOXM1, a member of the Fork head box (FOX) family, acts as an upstream regulator of SPDEF [25].

Furthermore, Shi et al. (2019) [26] identified the Runt-related transcription factor (RUNX2) as a factor that binds to the SPDEF promoter, thereby activating goblet cell differentiation (Figure 3). They also demonstrated that the knockdown of RUNX2 significantly decreased differentiation and mucus production after IL-13 exposure [26]. More recently, luciferase assays and chromatin immunoprecipitation PCR (ChIP-PCR) studies revealed that RUNX2 is a direct target of miR-30a-3p [28]. In vitro assays confirmed that the inhibition of miR-30a-3p expression enhanced IL-13-induced expression of RUNX2 and HMGB1 in airway epithelial cells [28].

Other studies conducted on primary human airway epithelial cells and airway cell lines have indicated that KLF5 acts as a repressor of CFTR transcript [22,110]. Interestingly, scRNAseq analysis revealed that CFTR transcripts were found in low abundance in many SCs, while high levels were restricted to rare pulmonary ionocytes [22]. Similarly, Sousa et al. (2020) [27] demonstrated that another member of the KLF family, specifically KLF4, suppresses CFTR expression through an AKT-mediated pathway in the human lung (Figure 3). Hence, both KLF4 and KLF5 play pivotal roles in airway epithelial differentiation by regulating CFTR expression.

## 6. Conclusions and Remarks

Several key conclusions and remarks can be made regarding the differentiation programs of BCs and the formation of MCCs and SCs in the airway epithelium. 

BCs are crucial for airway development and possess self-renewal capabilities, which are independent of NOTCH signaling. NOTCH signaling, however, plays a role in the differentiation of adult airway BCs. Different NOTCH receptors and ligands are expressed in human bronchial epithelial cells and determine whether lung epithelial cells develop into secretory or ciliated lineages. MCC differentiation can occur through NOTCH signaling inhibition in BCs or the trans-differentiation of SCs. 

Specific transcription factors collaborate to regulate the MCC differentiation process, which involves basal body biogenesis, migration, the docking of basal bodies, and the generation of motile cilia. SCs can differentiate from BCs, and NOTCH signaling is involved in generating multipotent suprabasal cells that can further differentiate into secretory or ciliated cells. NOTCH ligands are necessary for this differentiation process. And once again, additional transcription factors are involved in the regulation of gene expression and overall differentiation processes, highlighting the complexity that requires further research to fully understand the differentiation process from BCs.

## 7. Future Directions

There are several potential future directions for research on the differentiation programs of BCs and the formation of MCCs and SCs in the airway epithelium. Further investigation is needed to understand the precise mechanisms of NOTCH signaling in the differentiation of BCs and their subsequent fate toward secretory or ciliated lineages. This could involve studying the specific interactions between different NOTCH receptors, ligands, and downstream effectors in regulating cell fate determination. 

Also, further research can focus on unraveling the complex transcriptional networks involving these factors and identifying additional transcriptional regulators involved in these processes, for example, investigating BC heterogeneity. BCs have been identified as a key cell population involved in airway development and repair. However, it is important to explore the heterogeneity within the BC population and understand whether distinct subpopulations contribute differentially to MCC and SC differentiation. 

These studies could also investigate how environmental factors, such as air pollutants or microbial exposures, influence the differentiation programs of BCs and how the formation of MCCs and SCs could provide important insights into respiratory health and disease. Additionally, exploring epigenetic modifications that regulate gene expression during differentiation processes may uncover new mechanisms and potential therapeutic targets.

Overall, further research in these directions will enhance our understanding of the complex differentiation programs in the airway epithelium and may have implications for the development of therapies targeting respiratory diseases and conditions.

## Figures and Tables

**Figure 2 ijms-24-14789-f002:**
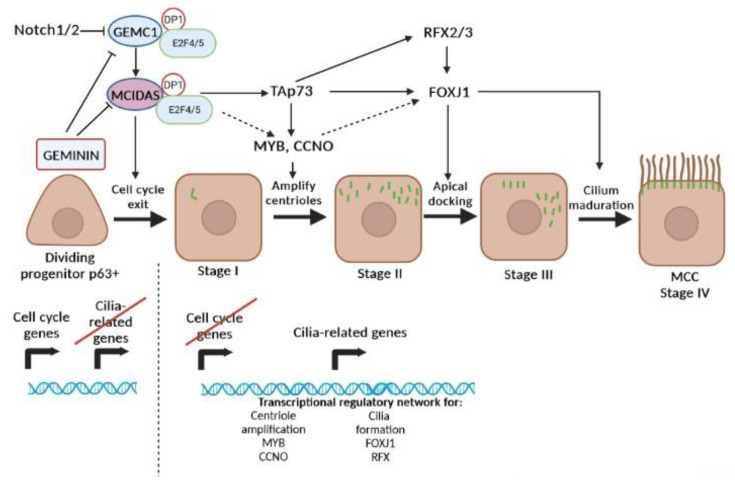
Molecular pathways involved in multiciliogenesis. Different transcriptional regulators associated with each step in the program are shown, along with the regulatory relationships among them, together with the NOTCH signaling pathway that controls the initiation of the program. Black lines ending with a perpendicular line indicate an inhibitory effect. Dotted arrows and black arrows illustrate stimulation by different transcription factors. In progenitor cells, GEMININ is expressed at high levels, inhibiting MCIDAS and GEMC1 [11]. At this point, cell cycle genes are active and cilia-related genes remain repressed. For cells to differentiate in multiciliate cell lineage, GEMININ levels must drop while the levels of GEMC1 increase. GEMC1 interact with the E2F4/5 and DP1 transcription factors, promoting the transcriptional activation of MCIDAS, which also forms complexes with E2F4/5 and DP1 [11]. These complexes activate genes required for centriole amplification and cilia formation, pushing cells toward multiciliogenesis. MCIDAS is upstream of TAp73, where TAp73 is a master transcriptional regulator of motile multiciliogenesis in airways, directly controlling MYB, CCNO, FOXJ1, and RFX2/3. MYB and CCNO are involved in centriole amplification, while FOXJ1 and RFX are implicated in cilia formation [17,31,74]. Created with BioRender.com (accessed on 10 July 2023).

**Figure 3 ijms-24-14789-f003:**
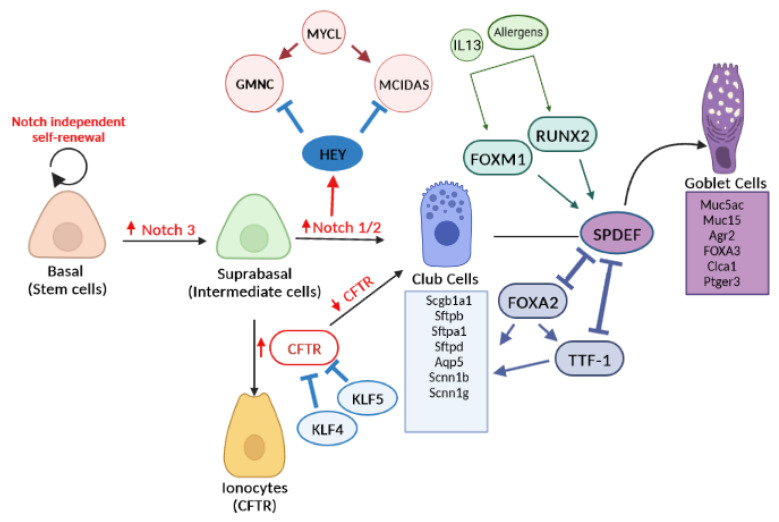
Schematic representation of secretory cell (SC) differentiation programs. Black lines ending with a perpendicular line indicate an inhibitory effect, while arrows show a stimulatory result; up- and downward red arrows indicate stimulation and suppression, respectively. Suprabasal cells activated by the NOTCH pathway activate the transcription factor HEY, leading to the repression of the MCC gene expression program and the differentiation in SCs [20]. Conversely, intermediate cells that do not receive active NOTCH signaling trigger a network involving MYCL, GMNC, and MCIDAS, leading to the differentiation into MCC precursors [20]. KLF4 and KLF5 repress the CFTR transcript, determining epithelial cell fate toward being secretory (club and goblet) [22,27]. The fates of club or goblet cells are determined by the expression of different transcription factors such as FOXM1, RUNX2, FOXA3, and SPDEF to goblet cells, while FOXA2 and TTF-1 inhibit genes implicated in goblet cell differentiation (Muc5ac, Muc15, Agr2, FOXA3, Clca1, Ptger3) [21,23,24,103]. In addition, FOXA2 and TTF-1 are relevant transcriptional regulators of genes expressed selectively in club cells (Scgb1a1, Sftpb, Sftpa1, Sftpd, Aqp5, Scnn1b, Scnn1g) [21,104,105,106]. Created with BioRender.com (accessed on 10 July 2023).

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
