# Peer review of "Transcriptional Regulation of Airway Epithelial Cell Differentiation: Insights into the Notch Pathway and Beyond"

_ijms, 2023, doi:10.3390/ijms241914789_

Round 1

Reviewer 1 Report

The article is consistent within itself. The references are relevant and recent. The cited sources are referenced correctly. Appropriate and key studies are included. The paper is comprehensive, the flow is logical and the data is presented critically.

However, there are some specific comments on weaknesses of the article and what could be improved:

Major points

1. There is a need of introduction, it could be 1/2 page long as well. There, present your background, the aim of the review, search strategy for the papers and how did you choose which to cite.

2. The structure of the paper is not comprehensive enough - try to make more paragraphs (every 6-7 rows), it`s hard to read the paper and the ideas. There are barely transitions between the paragraphs which further impede treading of the paper. Also, more subheadings are needed.

3. The review contains detailed information, however, there is no criticism, no discussion on the findings, the main line is missing - revise the text while focusing on the aim and main topic of your review. There should be logical connection between each paragraphs and subheadings.

Minor points

1. The first subheading - "the players" in what

2. How this heading "MCCs differentiation program" is related to the topic and title? Please, precise. This is valid for the rest of the headings as well.

3. Move the figures in the text right after their mentioning. The figures are great, though. 

The language is fine.

Author Response

POINT BY POINT RESPONSE TO REVIEWERS

 REVIEWER 1

Comments and Suggestions for Authors

The article is consistent within itself. The references are relevant and recent. The cited sources are referenced correctly. Appropriate and key studies are included. The paper is comprehensive, the flow is logical and the data is presented critically.

However, there are some specific comments on weaknesses of the article and what could be improved:

Major points

  1. There is a need of introduction, it could be 1/2 page long as well. There, present your background, the aim of the review, search strategy for the papers and how did you choose which to cite.

According to the reviewer’s suggestion, we have added an Introduction section to the manuscript, including the background and the aim of the review.

Regarding the search strategy to perform this review, we searched PubMed, Scopus, Web of Science and Google Scholar, usingdifferent keywords and the combination of each of them. On the one hand, the criteria for study inclusion were: (1) human and mice studies; (2) airway epithelial cells; (3) processes involved in airway ephitelium differentiation; (4) experimental in vitro models of cell differentiation; (5) publication year from 2004 to 2023; (6) English language; (7) papers with available abstract. On the other hand, exclusion criteria were: (1) studies in other species (e.g. zebrafish, xenopus…); (2) studies in other cellular models (e.g., skin cells, renal cells…); (3) studies focusing on epithelial cells development; (4) studies focusing on cancer cells and other respiratory pathology; (5) papers out of topic.

Two authors (G.C.L. and D.A.B.) independently screened the articles for relevance based on titles and abstracts. Then, the full texts of the retrieved studies were evaluated by all the authors. The management and organization of the bibliographic sources have been carried out using the computer program Mendeley (www.mendeley.com).  The extracted information and results were analyzed and summarized and, then it was included in this review.

  1. The structure of the paper is not comprehensive enough - try to make more paragraphs (every 6-7 rows), it`s hard to read the paper and the ideas. There are barely transitions between the paragraphs which further impede treading of the paper. Also, more subheadings are needed.

We have followed reviewer’s suggestion and the number of rows of the paragraphs has been reduced in the revised version of the manuscript. In addition, several subheadings have been added to the text.

  1. The review contains detailed information, however, there is no criticism, no discussion on the findings, the main line is missing - revise the text while focusing on the aim and main topic of your review. There should be logical connection between each paragraphs and subheadings.

According to this suggestion, we have included new subheadings throughout the manuscript and we have connected these several new subsections.

Minor points

  1. The first subheading - "the players" in what

We thank your suggestion. We have decided to eliminate this term.

  1. How this heading "MCCs differentiation program" is related to the topic and title? Please, precise. This is valid for the rest of the headings as well.

Following the suggestion, we have changed the heading in order to be more precise.

  1. Move the figures in the text right after their mentioning. The figures are great, though. 

We thank your suggestion. We have moved the figures.

Reviewer 2 Report

The authors discussed the Notch pathway contributes to regulation of airway epithelial cell differentiation. Three main cells are introduced, and their roles are also discussed. Furthermore, the authors also discussed the newly found cells in the epithelial layers. The authors also highlight the further research direction regarding the involvement of Notch pathway in airway epithelial cell differentiation. The manuscript was well written, I have some minor considerations as listed below:

1. Three figures are presented with lots of useful information. It will be much better for readers to understand the meaning of the figures if the authors clarified the meanings of all kinds of icons such as ↑, ↓, Ͱ in the figure legends.

2. Line 23, ";" should be added after transcription factors in keywords.

3. Line 262, citation of certain reference should be " Wang et al., 2015" but not "W. Wang et al., 2015). Similar things are also found in lines 320, 381, and 383. They should be revised during your resubmission.

4. COVID-19 virus is also affected airway epithelial cell layers, whether Notch pathway is involved in this process, this should be discussed in further studies.

Author Response

POINT BY POINT RESPONSE TO REVIEWERS

REVIEWER 2

Comments and Suggestions for Authors

The authors discussed the Notch pathway contributes to regulation of airway epithelial cell differentiation. Three main cells are introduced, and their roles are also discussed. Furthermore, the authors also discussed the newly found cells in the epithelial layers. The authors also highlight the further research direction regarding the involvement of Notch pathway in airway epithelial cell differentiation. The manuscript was well written, I have some minor considerations as listed below:

  1. Three figures are presented with lots of useful information. It will be much better for readers to understand the meaning of the figures if the authors clarified the meanings of all kinds of icons such as ↑, ↓, Ͱ in the figure legends.

 According to the reviewer’s suggestion, we have clarified the meanings of all kinds of icons

  1. Line 23, ";" should be added after transcription factors in keywords.

We thank your suggestion. We have added the ";" after transcription factors in keywords .

  1. Line 262, citation of certain reference should be " Wang et al., 2015" but not "W. Wang et al., 2015). Similar things are also found in lines 320, 381, and 383. They should be revised during your resubmission.

The citations have been corrected.

  1. COVID-19 virus is also affected airway epithelial cell layers, whether Notch pathway is involved in this process, this should be discussed in further studies.

Following the reviewer’s suggestion, we have added the following discussion to the subsection 3.1:

“Recently, up-regulation of NOTCH signaling was observed in SARS-CoV-2-infected cells (Rosa et al., 2021). Cells directly infected by SARS-CoV-2, NOTCH pathway might promote viral entry and in addition excessive NOTCH signaling may promote IL-6 and inflammatory pathways that can exacerbate morbidity and severity of COVID-19 (Baindara et al., 2022).”

Round 2

Reviewer 1 Report

The authors addressed all the issues and the paper has been improved significantly. The search strategy is perfect, please, add it into the text to bi available for the readers as well.

Author Response

We thank the reviewer for his/her comment. Following his/her suggestion, we have added our search criteria to write this manuscript in the main text.